# An Effective Strategy to Synthesize Well-Designed Activated Carbon Derived from Coal-Based Carbon Dots via Oxidation before Activation with a Low KOH Content as Supercapacitor Electrodes

**DOI:** 10.3390/nano13222909

**Published:** 2023-11-07

**Authors:** Yaojie Zhang, Jianbo Jia, Yue Sun, Bing Xu, Zhendong Jiang, Xiaoxiao Qu, Chuanxiang Zhang

**Affiliations:** College of Chemistry and Chemical Engineering, Henan Polytechnic University, Jiaozuo 454003, China; 13603449015@163.com (Y.Z.); jiajinabo@hpu.edu.cn (J.J.); 18137665111@163.com (Y.S.); xubinghpu@163.com (B.X.); 18623855902@163.com (Z.J.)

**Keywords:** activated carbons, KOH, coal-based carbon dots, supercapacitors

## Abstract

The development of coal-based activated carbon for supercapacitors provides a robust and effective approach toward the clean and efficient use of coal, and it also offers high-quality and low-cost raw materials for energy storage devices. However, the one-step activation method for preparing coal-based activated carbon has problems, such as difficulty in introducing surface-functional groups and high KOH dosage. In our work, activated carbon was prepared through an effective strategy of oxidation and KOH activation with a low KOH content by employing coal-based carbon dots as raw material. The influence of temperature during the KOH activation of carbon dots on a specific surface area, pore structure, and various quantities and types of surface-functional groups, as well as on the electrochemical performance of supercapacitors, was systematically studied. The as-prepared sample, with the alkali–carbon ratio of 0.75, processes a large specific surface area (1207 m^2^ g^−1^) and abundant surface-functional groups, which may provide enormous active sites and high wettability, thus bringing in high specific capacitance and boosted electrochemical performances. The oxygen and nitrogen content of the activated carbon decreases while the carbon content increases, and the activation temperature also increases. The as-prepared activated carbon reaches the highest specific capacitance of 202.2 F g^−1^ in a 6 M KOH electrolyte at a current density of 10 A g^−1^. This study provides new insight into the design of high-performance activated carbon and new avenues for the application of coal-based carbon dots.

## 1. Introduction

The development of clean energy can alleviate resource pressure, maintain ecological balance, and ensure the sustainable development of the economy [1,2]. However, its volatility and instability have produced a higher demand for efficient energy storage equipment [3]. Supercapacitors have been widely used in new energy vehicles, rail transit, and other fields in order to provide a fast charging and discharging speed, long cycle life, and high power density [4,5,6,7,8,9,10]. The electrode materials as a major influencing factor for supercapacitors have drawn extensive attention from related field researchers [11,12]. Electrode materials usually require good conductivity and high specific surface area, which indicates high specific capacitance and boosted electrochemical performances [13]. It is worth mentioning that activated carbon with a well-developed porous structure and low cost perfectly meets the above requirements [14]. Coal as a carbon material dominated by aromatic carbon has a molecular structure similar to that of activated carbon, indicating that it is a potential precursor to high-quality activated carbon materials [15,16,17,18,19,20]. Chemical activation, especially KOH activation, is the most commonly used and effective method to prepare activated carbon [21,22,23,24].

As early as 2009, Zhang et al. prepared activated carbon by employing KOH as the activator and anthracite as the raw material, and they applied it as a supercapacitor electrode, where the specific capacitance reached 264 F g^−1^ at 2 A g^−1^ with an alkali–carbon ratio of 4 [25]. In order to obtain activated carbon with better electrochemical performance, Shi et al. ground anthracite, using ball milling, to less than 10 μm before activation. When the alkali–carbon ratio was 4, the prepared activated carbon had the largest specific surface area, and the specific capacitance exceeded 300 F g^−1^ at the current density of 2 A g^−1^ [11]. However, some researchers believed that this solid-state activation method showed a low utilization of KOH. Therefore, Yue et al. proposed the impregnation method for preparing activated carbon with a reduced alkali–carbon ratio [26]. In addition, some researchers have also prepared high-performance activated carbon using low metamorphic coal or derivatives. For example, Xing et al. obtained activated carbon using lignite as raw material under the condition of an alkali–carbon ratio of 4, in which the specific capacitance reached 355 F g^−1^ at a current density of 0.05 A g^−1^, and the capacity retention rate was 93.9% after cycling 2000 times at a current density of 0.5 A g^−1^ [27]. Wang et al. used coal tar pitch as the precursor to produce activated carbon with a specific surface area of 2722 m^2^ g^−1^ and a specific capacitance of 295 F g^−1^ at a current density of 0.2 A g^−1^ at the same conditions [28].

However, in the above studies, there is still a high alkali–carbon ratio, that is, a large amount of KOH, which increased production costs as well as caused corrosion to the equipment. In traditional KOH activation, when the amount of KOH is low, only micropores can be generated in the carbon material, which are not conducive to the storage of electrolyte ions; only excess KOH can continue to etch carbon atoms on the micropore wall, thus expanding the pore size distribution [29]. Therefore, to prepare activated carbon with a certain mesoporous ratio and a suitable pore size distribution, a large amount of KOH is required [30,31]. In addition, KOH activation does not easily produce surface-functional groups that can generate pseudocapacitance [16].

Herein, we propose a viewpoint that can be used for the conversion of coal into coal-based carbon dots through potassium ferrate (K_2_FeO_4_) and hydrogen peroxide (H_2_O_2_). Carbon dots possess an extremely low ash content compared with that of coal, and their nano size and active site, produced by rich surface-functional groups, can also present a synergistic effect with improved reaction activity and reduced activation difficulty. As a result, carbon dots can be activated with KOH at a low alkali–carbon ratio to prepare well-developed porous activated carbon. Meanwhile, rich surface-functional groups can be retained on the surface of activated carbon after high-temperature activation, verifying the existence of pseudocapacitance and increasing the specific capacitance. The effect of temperature on the surface-functional groups, specific surface area, and supercapacitor performances of activated carbon are investigated systematically. The activated carbon thermally treated at 800 °C reaches the highest specific capacitance of 202.2 F g^−1^ in a 6 M KOH electrolyte with a current density of 10 A g^−1^. This work provides an efficient strategy to synthesize interconnected activated carbon derived from low-cost coal with low activator content as promising electrodes for high-performance supercapacitors and other energy storage systems.

## 2. Materials and Methods

### 2.1. Synthesis of CDAC-X

The synthesis of carbon dots from anthracite involved a two-step oxidation method, as previously described [32]. Then, 0.8 g of carbon dots were mixed with 0.6 g of KOH (the alkali–carbon ratio was 0.75). Subsequently, a mixture of 0.8 g of carbon dots and 0.6 g of KOH (with an alkali–carbon ratio of 0.75) was prepared. This blend was then placed in a tube furnace under a nitrogen atmosphere and incrementally heated at a rate of 5 °C/min until it reached temperatures of 600, 700, 800, and 900 °C, respectively. After maintaining these temperatures for a duration of 2 h, the resulting product was cooled to room temperature and sequentially washed with diluted hydrochloric acid and deionized water to achieve neutralization. Finally, vacuum drying was employed to obtain activated carbon. The activated carbon produced was labeled as CDAC-X (where X represents the respective carbonization temperature: 600, 700, 800, or 900).

### 2.2. Characterizations of CDAC-X

The morphology of CDACs was characterized using scanning electron microscopy (JSM-6390LV, Japan Elektronics Co., Ltd, Showashima, Japan). The measurements of the specific surface area and pore structure of CDACs were performed with N_2_ adsorption and desorption measurements (BELSORP Max II, MicrotracBEL Japan, Inc., Osaka, Japan). X-ray powder diffraction patterns and Raman spectra were recorded on X-ray diffraction (Smart Lab, Rigaku Corporation, Tokyo, Japan) with Cu Kα radiation (λ = 1.5406 Å) and Raman spectroscopy (Therno DXR2xi, Thermo Fisher Scientific, Waltham, MA, USA) with a 532 nm laser, respectively, to investigate the phase structure and crystal structure of CDACs. X-ray photoelectron spectra were recorded on X-ray photoelectron spectroscopy (ThermolFisher Nexsa, Thermo Fisher Scientific, MA, USA) to examine the surface elemental composition.

### 2.3. Electrochemical Measurements

The electrochemical performance of the CDAC-X measurements was evaluated using an electrochemical workstation (CHI760E, Shanghai, China) in a three-electrode system. The CDAC-X power, conductive carbon black (BP2000), and binder (PTFE) were combined in an 80:10:10 mass ratio. This mixture was then compressed into a thin film to create a working electrode. Circular sheets with a diameter of 0.5 cm were cut from the thin film and dried at 120 °C for 4 h under vacuum conditions. The electrode sheets were pressed onto a foam nickel measuring 1 × 3 cm^2^, with a pressure of 10 MPa, to form the working electrode. A Pt, foil measuring 1 × 1 cm^2^ served as the counter electrode, while an Hg/HgO electrode functioned as the reference electrode. The electrolyte used was a 6 M KOH solution.

The specific capacitance of the electrodes (C) was calculated based on the GCD curve using the following equation:C = I∆t/m∆V.
where C represents specific capacitance (F g^−1^), I denotes the discharging current (A), ∆t signifies the discharging time (s), m indicates the mass of the active material (g), and ∆V represents the voltage window range (V).

## 3. Results and Discussion

The pore structure of CDAC-X is analyzed by an N_2_ adsorption–desorption instrument, and the results are presented in Figure 1a. The adsorption isotherms of four kinds of activated carbon exhibited type Ⅳ, which is commonly observed in mesoporous carbon materials [33,34]. The entire process can be divided into three stages. In the first stage (P/P_0_ < 0.1), there is a rapid increase in N_2_ adsorption capacity, indicating the presence of micropores within the carbon materials. The second stage (0.1 < P/P_0_ < 0.9) represents a slower growth rate of N_2_ adsorption and corresponds to mesopores, where hysteresis loops are observed [35]. It should be noted that CDAC-900 exhibits the highest adsorption value in the first stage and has the smallest hysteresis loop in the second stage, suggesting a lower proportion of mesopores compared to other activated carbon materials [36,37]. The third stage is the saturated adsorption zone. The pore structure distribution (Figure 1b and inset) is aligned with the isothermal adsorption line: CDAC-900 primarily contains micropores ranging from 0.5 to 2 nm, while CDAC-600, CDAC-700, and CDAC-800 mainly consist of mesopores ranging from 2 to 15 nm along with some percentage of microporous pores as well. Furthermore, for CDAC-600 specifically, its mesopore distribution predominantly falls within a range of 7.5 to 12.5 nm; for CDAC-700, it ranges from 5 to 10 nm; and for CDAC-800, it ranges from 2 to 5 nm. Generally speaking, an increase in temperature can lead to more concentrated pore size distributions.

The specific surface area and pore structure parameters of CDAC-X are presented in Table 1. The specific surface area of CDAC-600 and CDAC-700 is comparatively lower, at 439 m^2^ g^−1^ and 668 m^2^ g^−1^, respectively. However, the specific surface area significantly increases to a remarkable value of 1207 m^2^ g^−1^ for CDAC-800. Furthermore, it can be observed that both microporous and mesoporous volumes exhibit an upward trend with increasing temperature, indicating that higher temperatures contribute to pore development. Nonetheless, upon reaching an activation temperature of 900 °C, although there is a slight increase in the specific surface area of CDAC-900, both total pore volume and the proportion of mesopores decrease. This phenomenon can be attributed to the complete activation of the carbon dots’ active sites at 900 °C, while KOH remains in a lower state, thereby favoring dominant pore formation. In summary, CDAC-800 not only demonstrates a significantly higher specific surface area (1207 m^2^ g^−1^) but also exhibits an appropriate distribution of pore sizes (microporous rate: 39.5%, mesoporous rate: 51.8%). These properties facilitate the adsorption of electrolyte ions for double electric layer formation, enhancing the utilization efficiency of the surface area and ultimately improving electrochemical performance during the charging/discharging processes [38].

The morphology of activated carbon from coal-based carbon dots obtained at different activation temperatures is shown in Figure 2. With the increase in the activation temperature, abundant pores gradually appear on the surface of the activated carbon, and the pores continue to expand and deepen, forming numerous uniformly distributed honeycomb pores. When the activation temperature is 600 °C, the porous structure is formed initially (Figure 2a); with the further increase in the activation temperature, the activation reaction is strengthened and more pores are formed (Figure 2b); when the activation temperature is 800 °C, the activation reaction is more intense, and the number of pores of activated carbon increases sharply (Figure 2c). Therefore, CDAC-800 has a porous structure and high specific surface area, which is conducive to charge storage and thus enhances the specific capacitance. Moreover, CDAC-900 in Figure 2d has negligible changes in morphology compared with CDAC-800.

The XRD pattern of CDAC-X is shown in Figure 3a. For activated carbon prepared at different activation temperatures, two diffraction peaks are observed near 24.7° and 43.6°, corresponding to the (002) and (100) planes of graphite crystal, respectively. With an increase in activation temperature up to 900 °C, the sharpness of these peaks diminishes, and their intensity weakens, indicating a decrease in graphitization degree and an enhancement of amorphous structure. This can be attributed to the intensified higher temperature leading to the insertion of KOH into carbon sites and subsequent reactions with carbon atoms that result in disruption of graphite microcrystalline structure [37]. Among all samples, CDAC-800 exhibits the sharpest characteristic peak, suggesting superior crystallinity which is advantageous for enhancing its rate performance [24,31].

Figure 3b demonstrates that the Raman spectrum of CDAC-X exhibits a D-band (1350 cm^−1^) and a G-band (1586 cm^−1^), which are respectively associated with sp^3^ defects and sp^2^ carbon. The I_D_/I_G_ ratio serves as an indicator for the degree of disorder in carbon-based materials. With the increase in activation temperature, the I_D_/I_G_ ratios for CDAC-600, CDAC-700, CDAC-800, and CDAC-900 incrementally rise from 0.665 to 0.895. This finding suggests that activated carbon materials with higher activation temperatures possess a greater number of defective sites due to the formation of pores, which disrupts the graphite lattice and reduces orderliness [24,39].

The X-ray photoelectron spectroscopy (XPS) method was utilized to examine the surface composition of CDAC-X. The experimental results revealed that carbon dots derived from coal-based materials and activated at varying temperatures incorporated carbon, oxygen, and nitrogen elements (Figure 4). By analyzing the C 1s spectrum peaks, as shown in Figure 5, it was determined that five distinct carbon atom types were present on the activated carbon surface: C=C/C–C (284.7 eV), C–N (285.4 eV), C–O (285.9 eV), C=O (286.4 eV), and O–C=O (287.2 eV) [31,40]. The presence of oxygen-containing functional groups enhances surface wettability and increases pseudo-capacitance, thereby leading to an improvement in the electrochemical performance of activated carbon [41].

Table 2 presents the relative contents of elements and carbon-functional groups. As the activation temperature increases, there is an increase in carbon atom content, whereas a decrease in oxygen atom content indicates that higher temperatures facilitate the removal of oxygen-functional groups from activated carbon. The content of C=O remains relatively unchanged while the content of C–N and O–C=O shows a decreasing trend; on the other hand, the content of C=C/C–C and C–O gradually increases with increasing activation temperature.

The atomic content of nitrogen also decreases with the increase in the activation temperature, suggesting that KOH not only etches away the carbon atoms but also consumes nitrogen atoms. The high-resolution N 1s spectrum of CDAC-X (Figure 6) is fitted to three peaks, assigned to N–6 (398.6 eV), N–5 (400.4 eV), and N–Q (402.3 eV) [42,43]. Among these forms, N–6 and N–5 are the predominant types of nitrogen. These two forms provide conjugated electrons and sufficient active sites beneficial for enhancing the supercapacitor’s electrochemical performance [44].

By calculating the integral area of each fitting peak, it can be observed that the relative proportion of N–6 increases with the rise in activation temperature, while that of N–5 decreases. Furthermore, analyzing the overall content variation of N–containing functional groups concerning activation temperature (as shown in Figure 7), it can be inferred that the content of N–6 remains relatively stable, whereas both N–5 and N–Q gradually decrease. Notably, there is a more significant reduction in N–5 content. Hence, it can be concluded that the decline in nitrogen elements primarily stems from the removal of N–5 content. This phenomenon can be attributed to the lower thermal stability of pyrrole-type nitrogen (N–5) compared to pyridine-type nitrogen (N–6), as indicated by their respective activation energies [45]. Although a lower activation temperature is not favorable for pore development, it facilitates the retention of oxygen-containing and nitrogen-containing functional groups within the carbon dots themselves, which improves the wettability of material and increases pseudocapacitance.

The electrochemical performance of activated carbon was evaluated in a three-electrode system (6 M KOH). The CV curves of CDAC-X at a scan rate of 5 mV s^−1^ are depicted in Figure 8a. All the CV curves exhibit a slightly curved rectangular shape, showing the coexistence of double-layer capacitance and pseudocapacitance, accompanied by minor distortions attributed to oxidation-reduction reactions caused by oxygen and nitrogen impurity atoms on the surface of activated carbon [46]. Notably, the CV curve of CDAC-800 has the largest area, suggesting the highest specific capacitance. XPS analysis (Figure 5 and Figure 6) reveals that oxygen and nitrogen impurity atoms are corresponding to carboxyl and pyrrole nitrogen, respectively [47]. However, their contents decrease with increasing temperature. Notably, when activated at temperatures of 600 °C and 700 °C, these functional groups exhibit higher levels leading to significant pseudocapacitance effects. As the activation temperature rises to 800 °C and 900 °C, both functional group contents decrease significantly—resulting in less pronounced pseudocapacitance effects. The GCD curves present an isosceles triangular shape that is not very symmetrical, providing further evidence for the coexistence of pseudocapacitance and double layers. Moreover, the GCD curves of CDAC-800 demonstrate the longest discharge time, thereby confirming its excellent specific capacity in line with the conclusion obtained from CV analysis. Figure 9 shows the cyclic voltammetry curves at different scanning speeds and the constant current charge–discharge curves at different current densities of four activated carbon materials. As shown in Figure 9(a1 b1), because of the high oxygen content, CDAC-600 and CDAC-700 exhibit both electric double-layer capacitance behavior and a certain pseudo-capacitance behavior. Therefore, the CV curves will be deformed with the increased scanning rate, and the corresponding GCD curves at high current density are also deformed. However, the CV and GCD curves of CDAC-800 and CDAC-900 do not change significantly with the increase in scanning rate and current density and still have the characteristics of the electric double layer. Especially when the current density is 50 A g^−1^ or even 100 A g^−1^, the GCD curves can maintain the shape of an isosceles triangle.

According to the principle of electric double-layer energy storage, the energy storage capacity of supercapacitors increases with the augmentation of the specific surface area of electrode materials [29]. However, it cannot be seen that the specific capacitance of CDAC-800 and CDAC-900 at 0.5 A g^−1^ current density conforms to this rule from Figure 8b. CDAC-900 exhibits the highest micropore rate and narrowest pore distribution. Consequently, electrolyte ions are unable to fully immerse in the electrode, resulting in the underutilization of many pores and a low effective utilization rate of surface area. Additionally, due to tiny pores and narrow ion diffusion channels in CDAC-900, ion transmission is limited, thereby hindering improvement in its specific capacitance. Although CDAC-800 has a smaller specific surface area compared to CDAC-900, it possesses a hierarchical porous structure consisting of 39.5% micropores, and 51.8% mesopores, respectively; thus, it exhibits a high utilization rate for its specific surface area, leading to a specific capacitance value of 243.6 F g^−1^ at a current density of 0.5 A g^−1^. Moreover, this structure facilitates a reduction in ion diffusion path length in the electrolytes, thereby minimizing diffusion resistance and enhancing transmission speed. As a result, the rate performance of the capacitor is ultimately improved [48]. Even at a current density of 10 A g^−1^, CDAC-800 maintains a specific capacitance of 202.2 F g^−1^. The specific capacitance and rate capability are better than the results reported in the previous literature (summarized in Table 3). Furthermore, when the current density is increased to 100 A g^−1^, CDAC-800 still exhibits a specific capacitance of 179.5 F g^−1^ with a capacity retention rate of 73.7%. The specific capacitances of CDAC-X at different current densities are shown in Figure 8c and Table 4.

Figure 8d displays the Nyquist plot spectrum of activated carbon, accompanied by an analog equivalent circuit diagram. The curve comprises three regions: high-frequency, medium-frequency, and low-frequency. In the high-frequency region, the intercept between the real axis (z’) and curve refers to the equivalent series resistance (Rs) of the test device, representing the sum of electrolyte ion resistance, internal resistance of electrode material, and contact resistance between working electrode and current collector. All of the Rs values are less than 1 Ω, indicating that the electrode has good conductivity. The semicircular ring appearing in the medium-frequency region is related to the resistance of charge transfer (Rct). The resistance value of CDAC-800 is the smallest, indicating that the electrode material and electrolyte interface can carry out ion transfer rapidly, which is mainly related to the appropriate hierarchical pore structure [11]. In the low-frequency region, the slope of the straight line corresponds to the capacitance characteristic. The impedance curves of CDAC-800 are nearly vertical straight lines, which shows the optimal double-layer capacitance behavior.

The evaluation of a supercapacitor’s electrochemical performance is contingent on numerous factors, with cyclic stability being particularly crucial. As illustrated in Figure 8e, the cyclic performance graph of CDAC-800 is presented. The CDAC-800 electrode demonstrates a specific capacitance of 191.6 F g^−1^ at a current density of 20 A g^−1^ and retains approximately 98.1% of its capacitance after undergoing 10,000 cycles, indicating its exceptional cycling stability. This exceptional cycling stability can be attributed to its high degree of crystallinity and hierarchical pore structure, which facilitate efficient charge transfer and ion diffusion [54]. 

## 4. Conclusions

In summary, coal-based carbon dots were utilized as precursors to the preparation of activated carbon at a low alkali–carbon ratio. The impact of temperature on the surface-functional groups, specific surface area, and supercapacitor performance of the activated carbon was systematically investigated. With the increase in temperature, the specific surface area of the activated carbon increased, and the total pore volume and mesopore volume first increased and then decreased. In addition, the oxygen content and nitrogen content of CDAC-X gradually decreased, especially carboxyl nitrogen and pyridine nitrogen. The specific surface area of the as-prepared sample reached 1207 m^2^ g^−1^ with an optimal pore size distribution consisting of micropores (39.5%) and mesopores (51.8%). Consequently, CDAC-800 exhibited superior electrochemical performance with a specific capacitance value of 243.6 F g^−1^ at a current density of 0.5 A g^−1^. These results indicate that utilizing coal-based carbon dots as precursors to prepare well-developed activated carbon through oxidation before activation not only simplifies the activation process but also introduces beneficial surface-functional groups in order to facilitate efficient and clean utilization of coal resources.

## Figures and Tables

**Figure 1 nanomaterials-13-02909-f001:**
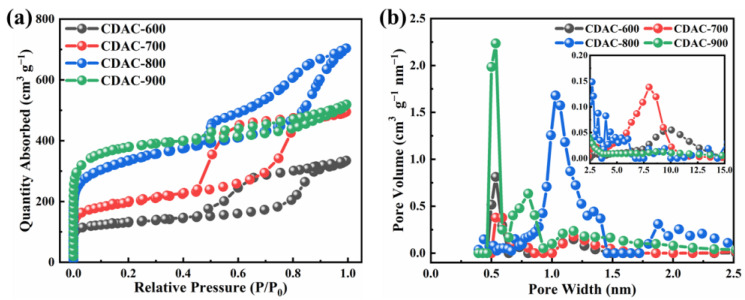
N_2_ adsorption–desorption isotherm (**a**) and pore size distribution curves (**b**) of CDAC-X. (Inset in (**b**): pore size distribution curves within the range of 2.5–15 nm).

**Figure 2 nanomaterials-13-02909-f002:**
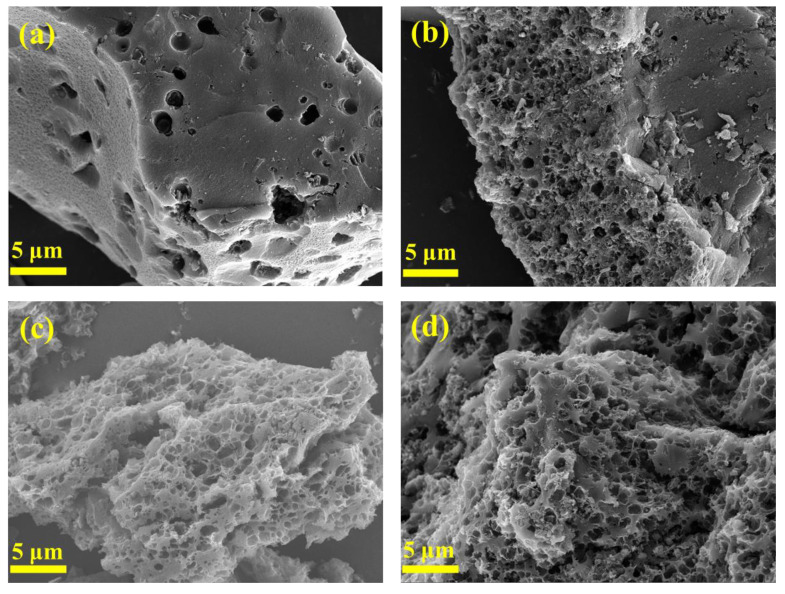
SEM images of CDAC-X: (**a**) CDAC-600; (**b**) CDAC-700; (**c**) CDAC-800; (**d**) CDAC-800.

**Figure 3 nanomaterials-13-02909-f003:**
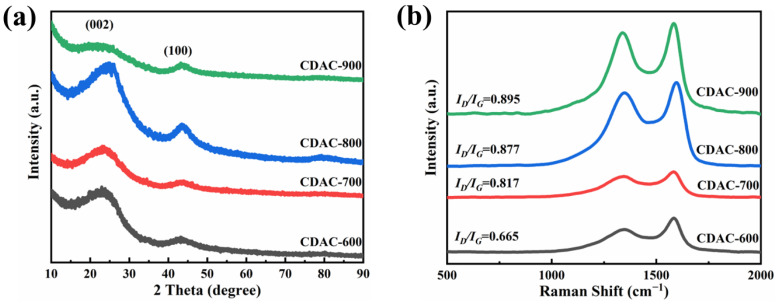
XRD patterns (**a**) and Raman spectra (**b**) of CDAC-X.

**Figure 4 nanomaterials-13-02909-f004:**
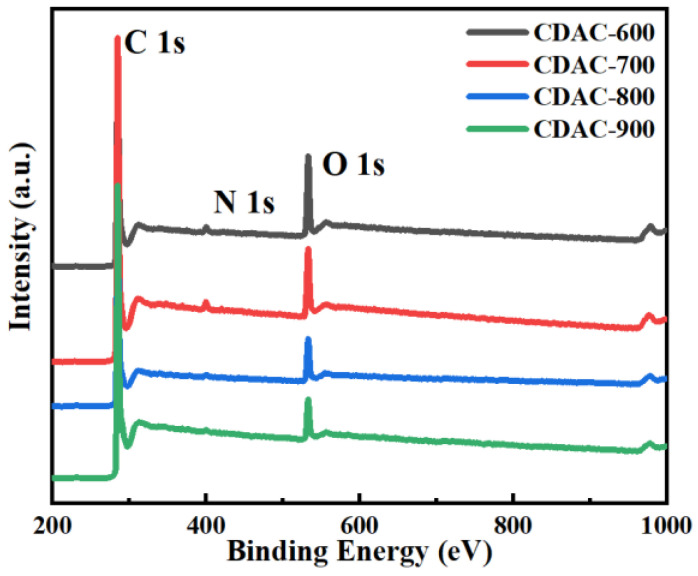
XPS survey spectra of CDAC-X.

**Figure 5 nanomaterials-13-02909-f005:**
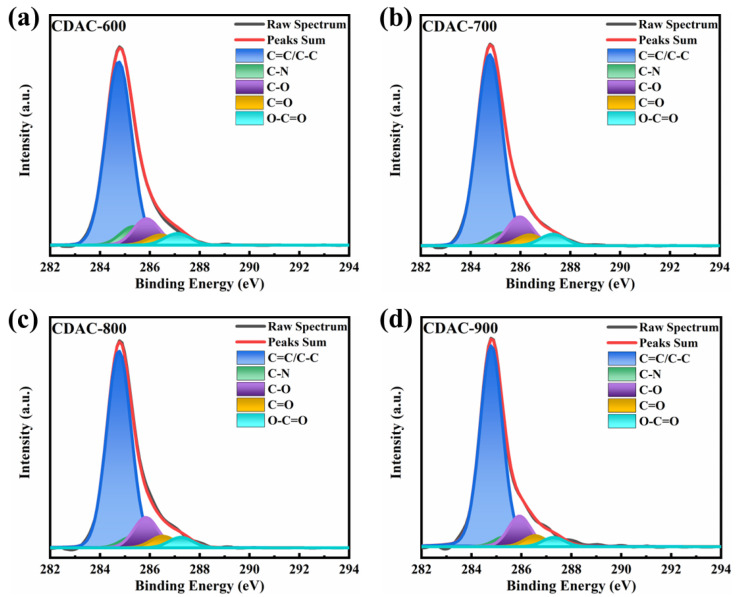
C 1s spectra with fitted peaks of CDAC-X: (**a**) CDAC-600; (**b**) CDAC-700; (**c**) CDAC-800; (**d**) CDAC-900.

**Figure 6 nanomaterials-13-02909-f006:**
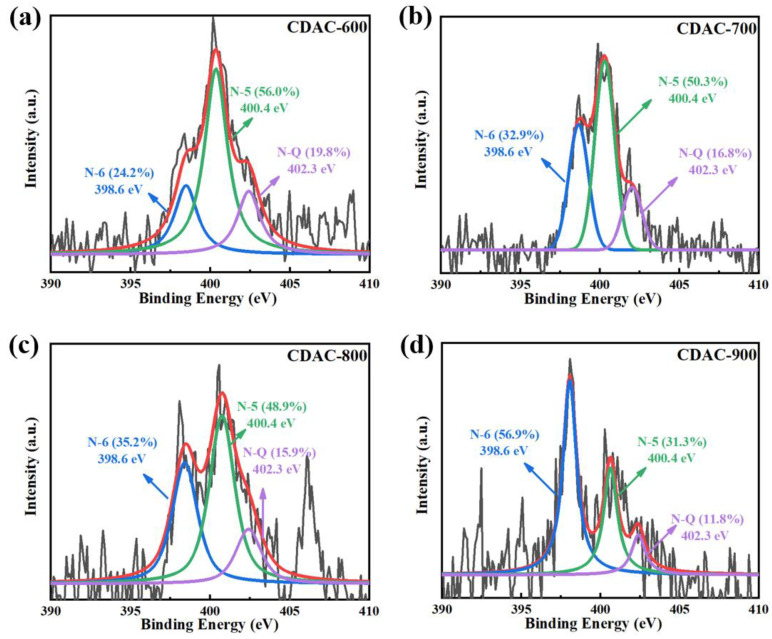
N 1s spectra with fitted peaks of CDAC-X: (**a**) CDAC-600; (**b**) CDAC-700; (**c**) CDAC-800; (**d**) CDAC-900.

**Figure 7 nanomaterials-13-02909-f007:**
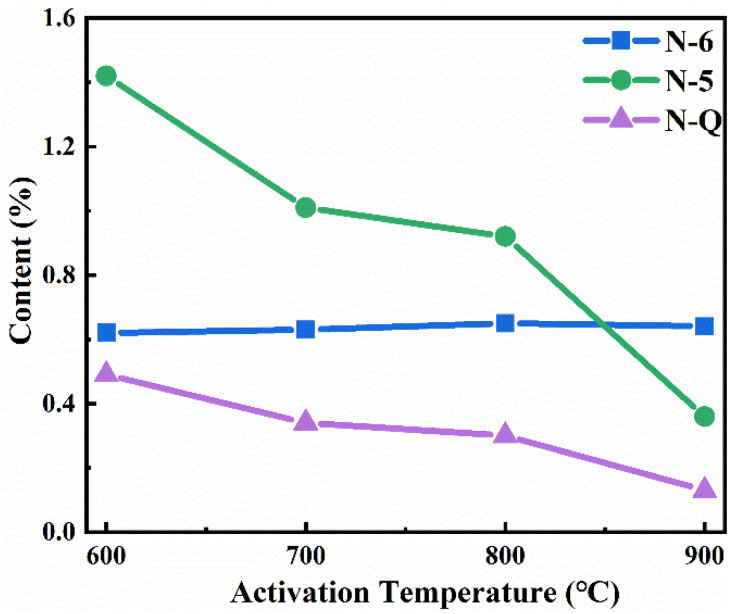
The variation trend of N-functional groups of CDAC-X.

**Figure 8 nanomaterials-13-02909-f008:**
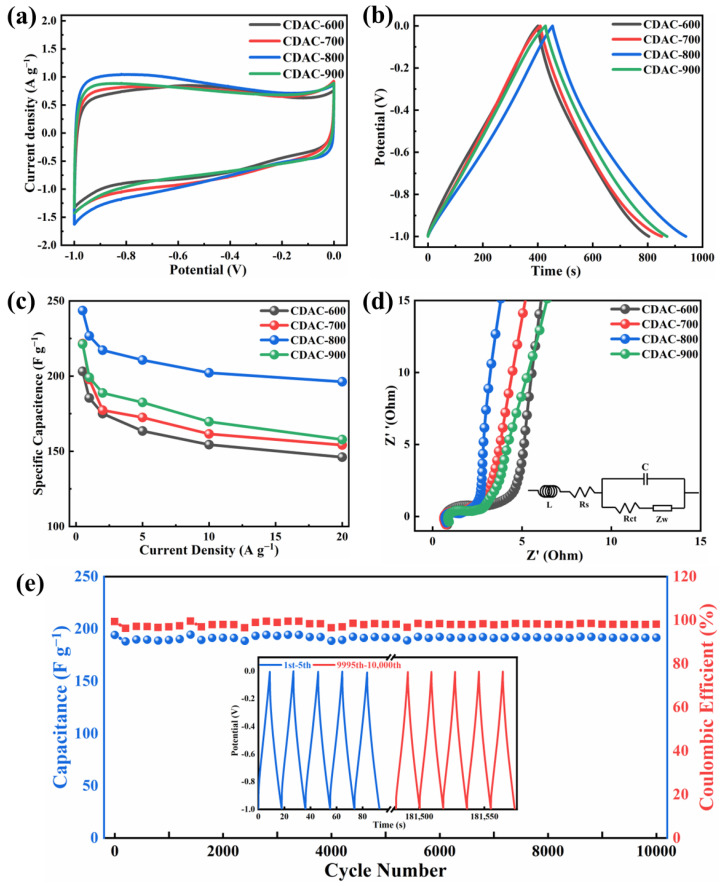
Electrochemical performances of CDAC-X: (**a**) CV curves of CDAC-X at 5 mV s^−1^; (**b**) GCD profiles of CDAC-X at 0.5 A g^−1^; (**c**) the specific capacitances of CDAC-X from 0.5 to 20 A g^−1^; (**d**) Nyquist plots; (**e**) The performance with 10,000 cycles of CDAC-800 at 20 A g^−1^. X is the carbonization temperature.

**Figure 9 nanomaterials-13-02909-f009:**
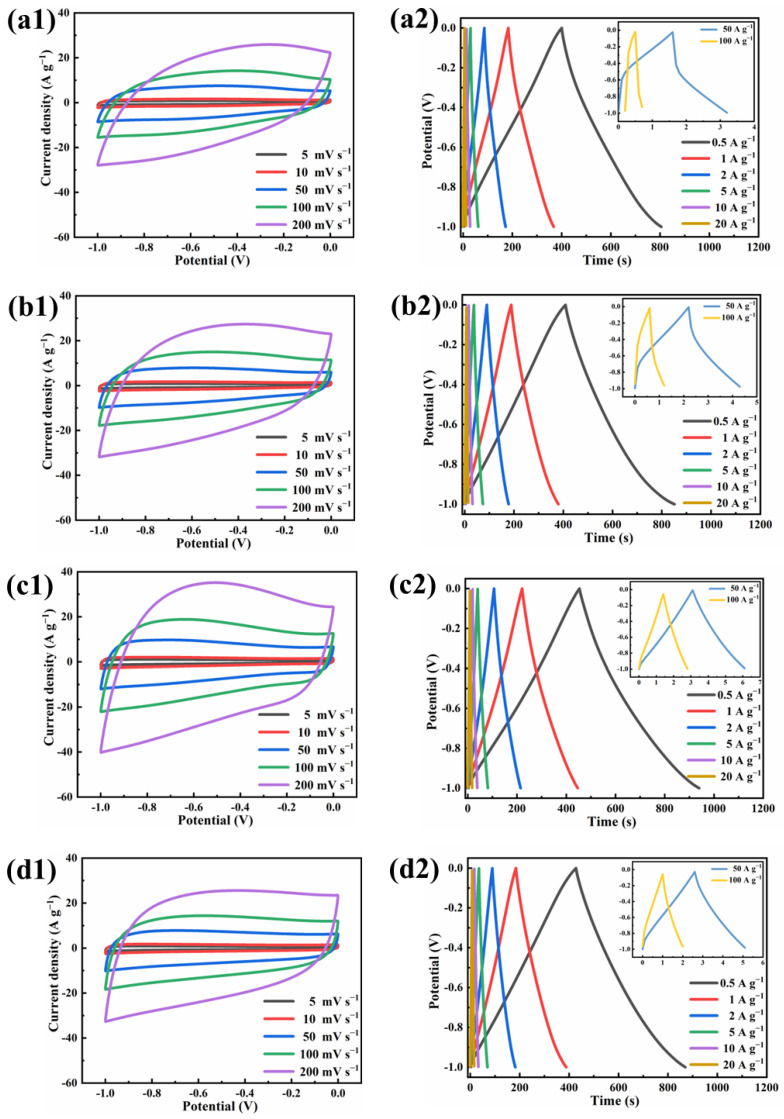
CV curves of CDAC-X measured from 5 to 200 mV s^−1^ and GCD curves of CDAC-X from 0.5 to 20 A g^−1^: (**a1**–**d1**) CV curves; (**a2**–**d2**) GCD curves. X is the activation temperature.

**Table 1 nanomaterials-13-02909-t001:** Specific surface area and pore structure parameters of CDAC-X.

Sample	S_BET_ (m^2^ g^−1^)	V_total_ (cm^3^ g^−1^)	V_mic_ (cm^3^ g^−1^)	V_mes_ (cm^3^ g^−1^)	V_mac_ (cm^3^ g^−1^)	V_mic_/V_total_ (%)	V_mes_/V_total_ (%)
CDAC-600	439	0.444	0.129	0.307	0.008	29.1	69.1
CDAC-700	668	0.671	0.172	0.497	0.002	25.6	74.1
CDAC-800	1207	1.346	0.531	0.697	0.118	39.5	51.8
CDAC-900	1260	0.662	0.453	0.199	0.010	68.4	30.0

**Table 2 nanomaterials-13-02909-t002:** Elemental and functional group content of CDAC-X.

Sample	C (at.%)						O (at.%)	N (at.%)
	Total C	C=C/C–C	C–N	C–O	C=O	O–C=O	Total O	Total N
CDAC-600	81.2	72.6	7.4	10.5	4.1	4.6	16.2	2.6
CDAC-700	84.4	74.8	5.2	11.3	4.2	4.5	13.6	2.0
CDAC-800	86.6	75.5	4.4	11.6	4.5	4.0	11.6	1.8
CDAC-900	91.1	76.5	3.8	11.7	4.2	3.5	7.8	1.1

**Table 3 nanomaterials-13-02909-t003:** The summary of specific capacitance for the different coal-based materials.

Precursor	Activation Method	Ratio of Activator to Precursor	Electrolyte	Specific Capacitance in Three-Electrode System	Reference
Anthracite	KOH	12 M solution	6 M KOH	199/1 A g^−1^	[26]
Coal tar pitch	KOH	4:1	6 M KOH	192/10 A g^−1^	[28]
Lignite	ZnCl_2_	3 M solution	6 M KOH	140/10 A g^−1^	[30]
Biomolecule based carbon dots	carbonization	-	1 M TBAPF_6_ in acetonitrile	6/100 A g^−1^	[49]
Bituminous	K_2_S	0.9:1	6 M KOH	123/10 A g^−1^	[50]
Coal	NH_3_	1:1	6 M KOH	129/50 A g^−1^	[51]
Oxidized coal	PAN	1:1	6 M KOH	175/100 A g^−1^	[52]
Carbon nanosheets	KOH	1:1	6 M KOH	170/100 A g^−1^	[53]
Coal-based carbon dots	KOH	0.75:1	6 M KOH	202/10 A g^−1^ 180/100 A g^−1^	This work

**Table 4 nanomaterials-13-02909-t004:** The specific capacitance of CDAC-X at different current densities.

Samples	Specific Capacitance (F g^−1^)					
	0.5 A g^−1^	1 A g^−1^	2 A g^−1^	5 A g^−1^	10 A g^−1^	20 A g^−1^
CDAC-600	203.1	186.5	175.0	163.5	154.4	146.1
CDAC-700	221.6	197.8	177.3	172.5	161.5	154.1
CDAC-800	243.6	226.7	217.2	210.7	202.2	196.2
CDAC-900	221.3	199.1	188.8	182.5	169.7	157.8

## Data Availability

Data will be made available on request.

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
