# Peer review of "An Effective Strategy to Synthesize Well-Designed Activated Carbon Derived from Coal-Based Carbon Dots via Oxidation before Activation with a Low KOH Content as Supercapacitor Electrodes"

_nanomaterials, 2023, doi:10.3390/nano13222909_

Round 1

Reviewer 1 Report

Comments and Suggestions for Authors

The work «A Facile and Effective Strategy to Synthesize Well-Designed Activated Carbon Derived from Coal-Based Carbon Dots via Oxidation before Activation with a Low KOH Content as Enhanced Supercapacitor Electrodes» is devoted actual and significant topic about synthesis of electrochemically active electrode material. The abstract briefly describes the motivation and results of the study. In the introduction, the authors describe well the problem being solved and the proposed ways for this. All synthesis methods contain a detailed description of the operations performed. The results obtained are presented in the form of figures of good quality and informative. The revealed regularities have an explanation.  

1. I recommend supplementing the manuscript with a table comparing the characteristics obtained in the work and literature data.

2. The Nyquist plot has an inductive section in the high-frequency region, so the equivalent circuit should be adjusted.

3. Describe the impedance in more detail and compare the data obtained from it.

4. If the slope of the Nyquist curves is discussed, then rearrange them with the same X- and Y-axis scales.

Author Response

Comment #1: I recommend supplementing the manuscript with a table comparing the characteristics obtained in the work and literature data.

Our response: Thank you for your comment. We have supplemented the manuscript with a table comparing the characteristics obtained in the work and literature data, as listed in Table 3.

Comment #2: The Nyquist plot has an inductive section in the high-frequency region, so the equivalent circuit should be adjusted.

Our response: We are grateful to the reviewer for correcting our mistake. We have made corresponding modification to the equivalent circuit diagram, which the inductive section was added in front of the resistance Rs on the main road.

Comment #3: Describe the impedance in more detail and compare the data obtained from it.

Our response: We appreciate the comment. We have described impedance in more detail and compared the data obtained.

Our modification to the manuscript: “In the high-frequency region, the intercept between the real axis (z') and curve refers to the equivalent series resistance (Rs) of the test device, representing the sum of electrolyte ion resistance, internal resistance of electrode material and contact resistance between working electrode and current collector. All of the Rs values are less than 1 Ω, indicating that the electrode has good conductivity. The semicircular ring appearing in the medium-frequency region is related to the resistance of charge transfer (Rct). The resistance value of CDAC-800 is the smallest, indicating that the electrode material and electrolyte interface can carry out ion transfer rapidly, which is mainly related to the appropriate hierarchical pore structure. In the low-frequency region, the slope of the straight line corresponds to the capacitance characteristic. The impedance curves of CDAC-800 are nearly vertical straight lines, which shows the optimal double-layer capacitance behavior.”

Comment #4: If the slope of the Nyquist curves is discussed, then rearrange them with the same X- and Y-axis scales.

Our response: We are grateful to the reviewer for correcting our mistake. We have rearranged the Nyquist curves with the same X- and Y-axis scales.

Reviewer 2 Report

Comments and Suggestions for Authors

It is a professionally executed and well-represented work on the elaboration of a new strategy to synthesize interconnected activated carbon derived from low-cost coal under low activator content as promising electrodes for high-performance supercapacitors and other energy storage systems. The authors were able to convincingly show that it is possible to create a good capacitor, utilizing coal-based carbon dots as precursors to prepare well-developed activated carbon through oxidation before activation. The suggested technology not only simplifies the activation process but also introduces beneficial surface functional groups to facilitate efficient and clean utilization of coal resources. This study provides new insight into the design of high-performance activated carbon and new avenues for the application of coal-based carbon dots. The paper meets the Nanomaterials’ requirements and can be recommended for publication in the journal in the present form.

Author Response

Our response: We thank the reviewer for this valuable comment.

Reviewer 3 Report

Comments and Suggestions for Authors

This is a good study with detailed structural characterization and electrochemical analyses. However, there are a few questions before it can be accepted.

1.      Why basic electrolyte was chosen for the study?

2.      Title should be revised to a simpler one.

3.      A Few important papers may be cited in the introduction: DOI:10.1016/j.apmt.2018.06.007; J. Mater. Chem. A, 2016,4, 16432-16445; DOI: 10.1002/eem2.12516.

4.      The performance should be compared with recent state of the arts carbons.

Comments on the Quality of English Language

Nothing major required.

Author Response

Comment #1: Why basic electrolyte was chosen for the study?

Our response: We thank the reviewer for carefully reviewing our work. Because aqueous electrolytes have advantages such as high ion concentration, high conductivity, low toxicity, and low cost, and can fully penetrate into the pores of electrode materials, they have been widely used. Compared to acidic electrolytes, basic electrolyte can also be used for pseudocapacitive materials such as metal oxides and sulfides. Moreover, which KOH owns enhanced current response compared to other electrolytes is due to the difference in the hydrated radius of K+ ion (3.31 Å), Na+ ions (3.58 Å), and Li+ ions (3.82 Å), respectively. It is noted that the reported conductivity of K+ (73 cm2/Ω mol) ions is greater than Na+ ions (50 cm2 /Ω mol) and Li+ ions (38 cm2 /Ω mol) at 25 ℃ and hence the mobility of ions would be higher for K+ compared with Na+ and Li+. Thus, we choose the 6 M KOH as the basic electrolyte for the study in our manuscript.

Comment #2: Title should be revised to a simpler one.

Our response: We thank the reviewer for this valuable comment. We have revised the title to a simple one, “An Effective Strategy to Synthesize Well-Designed Activated Carbon Derived from Coal-Based Carbon Dots with a Low KOH Content as Supercapacitor Electrodes”.

Our modification to the manuscript: “An Effective Strategy to Synthesize Well-Designed Activated Carbon Derived from Coal-Based Carbon Dots with a Low KOH Content as Supercapacitor Electrodes”.

Comment #3: A Few important papers may be cited in the introduction: DOI:10.1016/j.apmt.2018.06.007; J. Mater. Chem. A, 2016,4, 16432-16445; DOI: 10.1002/eem2.12516.

Our response: We thank the reviewer for this input. We reviewed the references that can guide the readers with future circumstances applicable to energy storage applications. We cited these suggested publications as the references for 12, 13, and 20 in the revised version.

Comment #4: The performance should be compared with recent state of the arts carbons.

Our response: We thank the reviewer for this input. The performance has been compared with the recent state of the arts carbons in Table 3.

Reviewer 4 Report

Comments and Suggestions for Authors

Manuscript Title: A Facile and Effective Strategy to Synthesize Well-Designed Activated Carbon Derived from Coal-Based Carbon Dots via Oxidation before Activation with a Low KOH Content as Enhanced Supercapacitor Electrodes

Ms. Ref. No.: nanomaterials-2673667 

In this manuscript, activated carbon was prepared through an effective strategy of oxidation and KOH activation with a low KOH content by employing coal-based carbon dots as raw material. The influence of temperature during KOH activation of carbon dots on specific surface area, pore structure, quantities and types of surface functional groups, and electrochemical performance of supercapacitors is systematically studied. After careful review, I believe the manuscript is suitable for publication in Nanomaterials. The primary concerns are outlined below for your reference:

1. Your manuscript suffers from considerable textural overlap with published manuscripts such as "Two-Step Hydrothermal Pretreatments for Co-Producing Xylooligosaccharides and Humic-like Acid from Vinegar Residue" and "Template-activated Bifunctional Soluble Salt ZnCl2 Assisted Synthesis of Coal-based Hierarchical Porous Carbon for High-performance Supercapacitors." The overall similarity was found to be 47%. Please address this issue.

2. The title of the manuscript should be more concise.

3. In Line 207, the caption for Figure 4 is incorrect. Please correct it.

4. Figure 5 should be referred to in the main text (lines 201-205).

5. Ensure consistent units throughout the manuscript, as there are variations between the figures and the main text.

6. Authors should provide an explanation of coulombic efficiency (Figure 8e) for better understanding. 

Please make these revisions, and your manuscript will be more polished and suitable for publication.

Author Response

Comment #1: Your manuscript suffers from considerable textural overlap with published manuscripts such as "Two-Step Hydrothermal Pretreatments for Co-Producing Xylooligosaccharides and Humic-like Acid from Vinegar Residue" and "Template-activated Bifunctional Soluble Salt ZnCl2 Assisted Synthesis of Coal-based Hierarchical Porous Carbon for High-performance Supercapacitors." The overall similarity was found to be 47%. Please address this issue.

Our response: We thank the reviewer for this insightful comment. We have reduced the repetition rate of individual articles to meet the standard.

Comment #2: The title of the manuscript should be more concise.

Our response: We thank the reviewer for this valuable comment. We have modified the title of the manuscript to a more concise one, “An Effective Strategy to Synthesize Well-Designed Activated Carbon Derived from Coal-Based Carbon Dots with a Low KOH Content as Supercapacitor Electrodes”.

Our modification to the manuscript: “An Effective Strategy to Synthesize Well-Designed Activated Carbon Derived from Coal-Based Carbon Dots with a Low KOH Content as Supercapacitor Electrodes”.

Comment #3: In Line 207, the caption for Figure 4 is incorrect. Please correct it.

Our response: We are grateful to the reviewer for correcting our mistake. We have changed the caption for Figure 4 to “XPS survey spectra of CDAC-X”.

Our modification to the manuscript: “Figure 4. XPS survey spectra of CDAC-X.”

Comment #4: Figure 5 should be referred to in the main text (lines 201-205).

Our response: We are grateful to the reviewer for pointing out our misleading information. We have referred to Figure 5 in the main text.

Our modification to the manuscript: “By analyzing the C 1s spectrum peaks, as shown in Figure 5, it was determined that five distinct carbon atom types were present on the activated carbon surface: C=C/C−C (284.7 eV), C−N (285.4 eV), C−O (285.9 eV), C=O (286.4 eV), and O−C=O (287.2 eV).”

Comment #5: Ensure consistent units throughout the manuscript, as there are variations between the figures and the main text.

Our response: We are grateful to the reviewer for correcting our mistake. We have changed the units in Figure 8 and Figure 9 to be consistent with the main text.

Comment #6: Authors should provide an explanation of coulombic efficiency (Figure 8e) for better understanding.

Our response: We thank the reviewer for this careful comment. We have explained the coulombic efficiency (Figure 8e).

Our modification to the manuscript: “The CDAC-800 electrode delivers a specific capacitance of 191.6 F g−1 at a current density of 20 A g−1 and maintains about 98.1% of the capacitance after 10000 cycles demonstrating its excellent cycling stability. This is due to its high crystallinity and hierarchical pore structure, which is conducive to charge transfer and ion diffusion.”